# Emergent Complexity and Zero-shot Transfer via Unsupervised Environment Design

**Michael Dennis**[*], **Natasha Jaques**[*2], **Eugene Vinitsky, Alexandre Bayen,**
**Stuart Russell, Andrew Critch, Sergey Levine**
University of California Berkeley AI Research (BAIR), Berkeley, CA, 94704
[2]Google Research, Brain team, Mountain View, CA, 94043
michael_dennis@berkeley.edu, natashajaques@google.com,
<evinitsky, bayen, russell, critch, svlevine>@berkeley.edu

## Abstract

A wide range of reinforcement learning (RL) problems — including robustness, transfer learning, unsupervised RL, and emergent complexity — require specifying a distribution of tasks or environments in which a policy will be trained. However, creating a useful distribution of environments is error prone, and takes a significant amount of developer time and effort. We propose Unsupervised Environment Design (UED) as an alternative paradigm, where developers provide environments with unknown parameters, and these parameters are used to automatically produce a distribution over valid, solvable environments. Existing approaches to automatically generating environments suffer from common failure modes: domain randomization cannot generate structure or adapt the difficulty of the environment to the agent's learning progress, and minimax adversarial training leads to worst-case environments that are often unsolvable. To generate structured, solvable environments for our *protagonist* agent, we introduce a second, *antagonist* agent that is allied with the *environment-generating adversary*. The adversary is motivated to generate environments which maximize regret, defined as the difference between the protagonist and antagonist agent's return. We call our technique Protagonist Antagonist Induced Regret Environment Design (PAIRED). Our experiments demonstrate that PAIRED produces a natural curriculum of increasingly complex environments, and PAIRED agents achieve higher zero-shot transfer performance when tested in highly novel environments.

## 1 Introduction

Many reinforcement learning problems require designing a distribution of tasks and environments that can be used to evaluate and train effective policies. This is true for a diverse array of methods including transfer learning (e.g., [2, 48, 32, 41]), robust RL (e.g., [4, 16, 28]), unsupervised RL (e.g., [15]), and emergent complexity (e.g., [38, 45, 46]). For example, suppose we wish to train a robot in simulation to pick up objects from a bin in a real-world warehouse. There are many possible configurations of objects, including objects being stacked on top of each other. We may not know *a priori* the typical arrangement of the objects, but can naturally describe the simulated environment as having a distribution over the object positions.

However, designing an appropriate distribution of environments is challenging. The real world is complicated, and correctly enumerating all of the edge cases relevant to an application could be impractical or impossible. Even if the developer of the RL method knew every edge case, specifying this distribution could take a significant amount of time and effort. We want to automate this process.

---

[*]Equal contribution

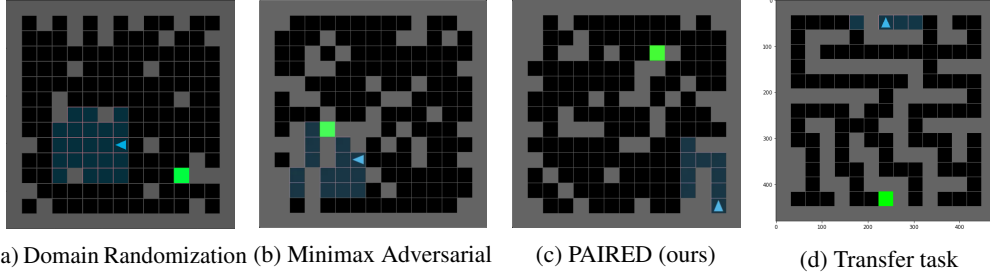

(a) Domain Randomization (b) Minimax Adversarial     (c) PAIRED (ours)     (d) Transfer task

Figure 1: An agent learns to navigate an environment where the position of the goal and obstacles is an underspecified parameter. If trained using domain randomization to randomly choose the obstacle locations (a), the agent will fail to generalize to a specific or complex configuration of obstacles, such as a maze (d). Minimax adversarial training encourages the adversary to create impossible environments, as shown in (b). In contrast, Protagonist Antagonist Induced Regret Environment Design (PAIRED), trains the adversary based on the difference between the reward of the agent (protagonist) and the reward of a second, antagonist agent. Because the two agents are learning together, the adversary is motivated to create a curriculum of difficult but achievable environments tailored to the agents' current level of performance (c). PAIRED facilitates learning complex behaviors and policies that perform well under zero-shot transfer to challenging new environments at test time.

In order for this automation to be useful, there must be a way of specifying the domain of environments in which the policy should be trained, without needing to fully specify the distribution. In our approach, developers only need to supply an *underspecified environment*: an environment which has free parameters which control its features and behavior. For instance, the developer could give a navigation environment in which the agent's objective is to navigate to the goal and the free parameters are the positions of obstacles. Our method will then construct distributions of environments by providing a distribution of settings of these free parameters; in this case, positions for those blocks. We call this problem of taking the underspecified environment and a policy, and producing an interesting distribution of fully specified environments in which that policy can be further trained *Unsupervised Environment Design* (UED). We formalize unsupervised environment design in Section 3, providing a characterization of the space of possible approaches which subsumes prior work. After training a policy in a distribution of environments generated by UED, we arrive at an updated policy and can use UED to generate more environments in which the updated policy can be trained. In this way, an approach for UED naturally gives rise to an approach for Unsupervised Curriculum Design (UCD). This method can be used to generate capable policies through increasingly complex environments targeted at the policy's current abilities.

Two prior approaches to UED are domain randomization, which generates fully specified environments uniformly randomly regardless of the current policy (e.g., [17, 32, 41]), and adversarially generating environments to minimize the reward of the current policy; *i.e.* minimax training (e.g., [28, 25, 43, 18]). While each of these approaches have their place, they can each fail to generate any interesting environments. In Figure 1 we show examples of maze navigation environments generated by each of these techniques. Uniformly random environments will often fail to generate interesting structures; in the maze example, it will be unlikely to generate walls (Figure 1a). On the other extreme, a minimax adversary is incentivized to make the environments completely unsolvable, generating mazes with unreachable goals (Figure 1b). In many environments, both of these methods fail to generate structured and solvable environments. We present a middle ground, generating environments which maximize regret, which produces difficult but solvable tasks (Figure 1c). Our results show that optimizing regret results in agents that are able to perform difficult transfer task (Figure 1d), which are not possible using the other two techniques.

We propose a novel adversarial training technique which naturally solves the problem of the adversary generating unsolvable environments by introducing an *antagonist* which is allied with the *environment-generating adversary*. For the sake of clarity, we refer to the primary agent we are trying to train as the *protagonist*. The environment adversary's goal is to design environments in which the antagonist achieves high reward and the protagonist receives low reward. If the adversary generates unsolvable environments, the antagonist and protagonist would perform the same and the adversary would get a score of zero, but if the adversary finds environments the antagonist solves and the protagonist does not solve, the adversary achieves a positive score. Thus, the environment adversary is incentivized to

create challenging but *feasible* environments, in which the antagonist can outperform the protagonist. Moreover, as the protagonist learns to solves the simple environments, the antagonist must generate more complex environments to make the protagonist fail, increasing the complexity of the generated tasks and leading to automatic curriculum generation.

We show that when both teams reach a Nash equilibrium, the generated environments have the highest regret, in which the performance of the protagonist's policy most differs from that of the optimal policy. Thus, we call our approach *Protagonist Antagonist Induced Regret Environment Design (PAIRED)*. Our results demonstrate that compared to domain randomization, minimax adversarial training, and population-based minimax training (as in Wang et al. [45]), PAIRED agents learn more complex behaviors, and have higher zero-shot transfer performance in challenging, novel environments.

In summary, our main contributions are: a) formalizing the problem of Unsupervised Environment Design (UED) (Section 3), b) introducing the PAIRED Algorithm (Section 4), c) demonstrating PAIRED leads to more complex behavior and more effective zero-shot transfer to new environments than existing approaches to environment design (Section 5), and d) characterizing solutions to UED and connecting the framework to classical decision theory (Section 3).

## 2   Related Work

When proposing an approach to UED we are essentially making a decision about which environments to prioritize during training, based on an underspecified environment set provided by the designer. Making decisions in situations of extreme uncertainty has been studied in decision theory under the name "decisions under ignorance" [27] and is in fact deeply connected to our work, as we detail in Section 3. Milnor [24] characterize several of these approaches; using Milnor's terminology, we see domain randomization [17] as "the principle of insufficient reason" proposed by Laplace, minimax adversarial training [18] as the "maximin principle" proposed by Wald [44], and our approach, PAIRED, as "minimax regret" proposed by Savage [33]. Savage's approach was built into a theory of making sequential decisions which minimize worst case regret [31, 19], which eventually developed into modern literature on minimizing worst case regret in bandit settings [6].

Multi-agent training has been proposed as a way to automatically generate curricula in RL [20, 21, 14]. Competition can drive the emergence of complex skills, even some that the developer did not anticipate [5, 13]. Asymmetric Self Play (ASP) [38] trains a demonstrator agent to complete the simplest possible task that a learner cannot yet complete, ensuring that the task is, in principle, solvable. In contrast, PAIRED is significantly more general because it generates complex new environments, rather than trajectories within an existing, limited environment. Campero et al. [7] study curriculum learning in a similar environment to the one under investigation in this paper. They use a teacher to propose goal locations to which the agent must navigate, and the teacher's reward is computed based on whether the agent takes more or less steps than a threshold value. To create a curriculum, the threshold is linearly increased over the course of training. POET [45, 46] uses a population of minimax (rather than minimax *regret*) adversaries to generate the terrain for a 2D walker. However, POET [45] requires generating many new environments, testing all agents within each one, and discarding environments based on a manually chosen reward threshold, which wastes a significant amount of computation. In contrast, our minimax regret approach automatically learns to tune the difficulty of the environment by comparing the protagonist and antagonist's scores. In addition, these works do not investigate whether adversarial environment generation can provide enhanced generalization when agents are transferred to new environments, as we do in this paper.

We evaluated PAIRED in terms of its ability to learn policies that transfer to unseen, hand-designed test environments. One approach to produce policies which can transfer between environments is unsupervised RL (e.g., [34, 36]). Gupta et al. [15] propose an unsupervised meta-RL technique that computes minimax regret against a diverse task distribution learned with the DIAYN algorithm [10]. However, this algorithm does not learn to modify the environment and does not adapt the task distribution based on the learning progress of the agent. PAIRED provides a curriculum of increasingly difficult environments and allows us to provide a theoretical characterization of when minimax regret should be preferred over other approaches.

The robust control literature has made use of both minimax and regret objectives. Minimax training has been used in the controls literature [37, 3, 49], the Robust MDP literature [4, 16, 26, 39], and

more recently, through the use of adversarial RL training [28, 43, 25]. Unlike our algorithm, minimax adversaries have no incentive to guide the learning progress of the agent and can make environments arbitrarily hard or unsolvable. Ghavamzadeh et al. [12] minimize the regret of a model-based policy against a safe baseline to ensure safe policy improvement. Regan and Boutilier [29, 30] study Markov Decision Processes (MDPs) in which the reward function is not fully specified, and use minimax regret as an objective to guide the elicitation of user preferences. While these approaches use similar theoretical techniques to ours, they use analytical solution methods that do not scale to the type of deep learning approach used in this paper, do not attempt to learn to generate environments, and do not consider automatic curriculum generation.

Domain randomization (DR) [17] is an alternative approach in which a designer specifies a set of parameters to which the policy should be robust [32, 41, 40]. These parameters are then randomly sampled for each episode, and a policy is trained that performs well on average across parameter values. However, this does not guarantee good performance on a specific, challenging configuration of parameters. While DR has had empirical success [1], it requires careful parameterization and does not automatically tailor generated environments to the current proficiency of the learning agent. Mehta et al. [23] propose to enhance DR by learning which parameter values lead to the biggest decrease in performance compared to a reference environment.

## 3 Unsupervised Environment Design

The goal of this work is to construct a policy that performs well across a large set of environments. We train policies by starting with an initially random policy, generating environments based on that policy to best suit its continued learning, train that policy in the generated environments, and repeat until the policy converges or we run out of resources. We then test the trained policies in a set of challenging transfer tasks not provided during training. In this section, we will focus on the environment generation step, which we call *unsupervised environment design* (UED). We will formally define UED as the problem of using an underspecified environment to produce a distribution over fully specified environments, which supports the continued learning of a particular policy. To do this, we must formally define fully specified and underspecified environments, and describe how to create a distribution of environments using UED. We will end this section by proposing minimax regret as a novel approach to UED.

We will model our fully specified environments with a Partially Observable Markov Decision Process (POMDP), which is a tuple $\langle A, O, S, \mathcal{T}, \mathcal{I}, \mathcal{R}, \gamma \rangle$ where $A$ is a set of actions, $O$ is a set of observations, $S$ is a set of states, $\mathcal{T} : S \times A \to \mathbf{\Delta}(S)$ is a transition function, $\mathcal{I} : S \to O$ is an observation (or inspection) function, $\mathcal{R} : S \to \mathbb{R}$, and $\gamma$ is a discount factor. We will define the utility as $U(\pi) = \sum_{i=0}^{T} r_t \gamma^t$, where $T$ is a horizon.

To model an underspecified environment, we propose the Underspecified Partially Observable Markov Decision Process (UPOMDP) as a tuple $\mathcal{M} = \langle A, O, \Theta, S^{\mathcal{M}}, \mathcal{T}^{\mathcal{M}}, \mathcal{I}^{\mathcal{M}}, \mathcal{R}^{\mathcal{M}}, \gamma \rangle$. The only difference between a POMDP and a UPOMDP is that a UPOMDP has a set $\Theta$ representing the free parameters of the environment, which can be chosen to be distinct at each time step and are incorporated into the transition function as $\mathcal{T}^{\mathcal{M}} : S \times A \times \Theta \to \mathbf{\Delta}(S)$. Thus a possible setting of the environment is given by some trajectory of environment parameters $\vec{\theta}$. As an example UPOMDP, consider a simulation of a robot in which $\vec{\theta}$ are additional forces which can be applied at each time step. A setting of the environment $\vec{\theta}$ can be naturally combined with the underspecified environment $\mathcal{M}$ to give a specific POMDP, which we will denote $\mathcal{M}_{\vec{\theta}}$.

In UED, we want to generate a distribution of environments in which to continue training a policy. We would like to curate this distribution of environments to best support the continued learning of a particular agent policy. As such, we can propose a solution to UED by specifying some *environment policy*, $\Lambda : \Pi \to \mathbf{\Delta}(\Theta^T)$ where $\Pi$ is the set of possible policies and $\Theta^T$ is the set of possible sequences of environment parameters. In Table 2, we see a few examples of possible choices for environment policies, as well as how they correspond to previous literature. Each of these choices also have a corresponding decision rule used for making *decisions under ignorance* [27]. Each decision rule can be understood as a method for choosing a policy given an underspecified environment. This connection between UED and decisions under ignorance extends further than these few decision rules. In the Appendix we will make this connection concrete and show that, under reasonable assumptions, UED and decisions under ignorance are solving the same problem.

Table 1: Example Environment Policies

| UED Technique | Environment Policy | Decision Rule |
|---|---|---|
| Domain Randomization [17, 32, 41] | $\Lambda^{DR}(\pi) = \mathcal{U}(\Theta^T)$ | Insufficient Reason |
| Minimax Adversary [28, 43, 25] | $\Lambda^M(\pi) = \underset{\vec{\theta} \in \Theta^T}{\mathrm{argmin}}\{U^{\vec{\theta}}(\pi)\}$ | Maximin |
| PAIRED (ours) | $\Lambda^{MR}(\pi) = \{\overline{\theta}_\pi : \frac{c_\pi}{v_\pi}, \tilde{D}_\pi : \text{otherwise}\}$ | Minimax Regret |

Table 2: The environment policies corresponding to the three techniques for UED which we study, along with their corresponding decision rule. Where $\mathcal{U}(X)$ is a uniform distribution over X, $\tilde{D}_\pi$ is a baseline distribution, $\overline{\theta}_\pi$ is the trajectory which maximizes regret of $\pi$, $v_\pi$ is the value above the baseline distribution that $\pi$ achieves on that trajectory, and $c_\pi$ is the negative of the worst-case regret of $\pi$, normalized so that $\frac{c_\pi}{v_\pi}$ is between 0 and 1. This is described in full in the appendix.

PAIRED can be seen as an approach for approximating the environment policy, $\Lambda^{MR}$, which corresponds to the decision rule minimax regret. One useful property of minimax regret, which is not true of domain randomization or minimax adversarial training, is that whenever a task has a sufficiently well-defined notion of success and failure it chooses policies which succeed. By minimizing the worst case difference between what is achievable and what it achieves, whenever there is a policy which ensures success, it will not fail where others could succeed.

**Theorem 1.** *Suppose that all achievable rewards fall into one of two class of outcomes labeled* **SUCCESS** *giving rewards in* $[\boldsymbol{S}_{min}, \boldsymbol{S}_{max}]$ *and* **FAILURE** *giving rewards in* $[\boldsymbol{F}_{min}, \boldsymbol{F}_{max}]$, *such that* $\boldsymbol{F}_{min} \leq \boldsymbol{F}_{max} < \boldsymbol{S}_{min} \leq \boldsymbol{S}_{max}$. *In addition assume that the range of possible rewards in either class is smaller than the difference between the classes so we have* $\boldsymbol{S}_{max} - \boldsymbol{S}_{min} < \boldsymbol{S}_{min} - \boldsymbol{F}_{max}$ *and* $\boldsymbol{F}_{max} - \boldsymbol{F}_{min} < \boldsymbol{S}_{min} - \boldsymbol{F}_{max}$. *Further suppose that there is a policy $\pi$ which succeeds on any $\vec{\theta}$ whenever success is possible. Then minimax regret will choose a policy which has that property.*

The proof of this property, and examples showing how minimax and domain randomization fail to have this property, are in the Appendix. In the next section we will formally introduce PAIRED as a method for approximating the minimax regret environment policy, $\Lambda^{MR}$.

# 4   Protagonist Antagonist Induced Regret Environment Design (PAIRED)

Here, we describe how to approximate minimax regret, and introduce our proposed algorithm, Protagonist Antagonist Induced Regret Environment Design (PAIRED). Regret is defined as the difference between the payoff obtained for a decision, and the optimal payoff that could have been obtained in the same setting with a different decision. In order to approximate regret, we use the difference between the payoffs of two agents acting under the same environment conditions. Assume we are given a fixed environment with parameters $\vec{\theta}$, a fixed policy for the protagonist agent, $\pi^P$, and we then train a second antagonist agent, $\pi^A$, to optimality in this environment. Then, the difference between the reward obtained by the antagonist, $U^{\vec{\theta}}(\pi^A)$, and the protagonist, $U^{\vec{\theta}}(\pi^P)$, is the regret:

$$\mathrm{REGRET}^{\vec{\theta}}\left(\pi^P, \pi^A\right) = U^{\vec{\theta}}\left(\pi^A\right) - U^{\vec{\theta}}\left(\pi^P\right) \tag{1}$$

In PAIRED, we introduce an environment adversary that learns to control the parameters of the environment, $\vec{\theta}$, to maximize the regret of the protagonist against the antagonist. For each training batch, the adversary generates the parameters of the environment, $\vec{\theta} \sim \tilde{\Lambda}$, which both agents will play. The adversary and antagonist are trained to maximize the regret as computed in Eq. 1, while the protagonist learns to minimize regret. This procedure is shown in Algorithm 1. The code for PAIRED and our experiments is available in open source at `https://github.com/google-research/google-research/tree/master/social_rl/`. We note that our approach is agnostic to the choice of RL technique used to optimize regret.

To improve the learning signal for the adversary, once the adversary creates an environment, both the protagonist and antagonist generate several trajectories within that same environment. This allows for a more accurate approximation the minimax regret as the difference between the maximum reward of the antagonist and the average reward of the protagonist over all trajectories: $\mathrm{REGRET} \approx$

$\max_{\tau^A} U^{\vec{\theta}}(\tau^A) - \mathbb{E}_{\tau^P}[U^{\vec{\theta}}(\tau^P)]$. We have found this reduces noise in the reward and more accurately rewards the adversary for building difficult but solvable environments.

---

**Algorithm 1:** PAIRED.

---

Randomly initialize Protagonist $\pi^P$, Antagonist $\pi^A$, and Adversary $\tilde{\Lambda}$;
**while** *not converged* **do**

> Use adversary to generate environment parameters: $\vec{\theta} \sim \tilde{\Lambda}$. Use to create POMDP $\mathcal{M}_{\vec{\theta}}$.
> Collect Protagonist trajectory $\tau^P$ in $\mathcal{M}_{\vec{\theta}}$. Compute: $U^{\vec{\theta}}(\pi^P) = \sum_{i=0}^{T} r_t \gamma^t$
> Collect Antagonist trajectory $\tau^A$ in $\mathcal{M}_{\vec{\theta}}$. Compute: $U^{\vec{\theta}}(\pi^A) = \sum_{i=0}^{T} r_t \gamma^t$
> Compute: $\text{REGRET}^{\vec{\theta}}(\pi^P, \pi^A) = U^{\vec{\theta}}(\pi^A) - U^{\vec{\theta}}(\pi^P)$
> Train Protagonist policy $\pi^P$ with RL update and reward $R(\tau^P) = -\text{REGRET}$
> Train Antagonist policy $\pi^A$ with RL update and reward $R(\tau^A) = \text{REGRET}$
> Train Adversary policy $\tilde{\Lambda}$ with RL update and reward $R(\tau^{\tilde{\Lambda}}) = \text{REGRET}$

**end**

---

At each training step, the environment adversary can be seen as solving a UED problem for the current protagonist, generating a curated set of environments in which it can learn. While the adversary is motivated to generate tasks beyond the protagonist's abilities, the regret can actually incentivize creating the easiest task on which the protagonist fails but the antagonist succeeds. This is because if the reward function contains any bonus for solving the task more efficiently (e.g. in fewer timesteps), the antagonist will get more reward if the task is easier. Thus, the adversary gets the most regret for proposing the easiest task which is outside the protagonist's skill range. Thus, regret incentivizes the adversary to propose tasks within the agent's "zone of proximal development" [8]. As the protagonist learns to solve the simple tasks, the adversary is forced to find harder tasks to achieve positive reward, increasing the complexity of the generated tasks and leading to automatic curriculum generation.

Though multi-agent learning may not always converge [22], we can show that, if each team in this game finds an optimal solution, the protagonist would be playing a minimax regret policy.

**Theorem 2.** *Let $(\pi^P, \pi^A, \vec{\theta})$ be in Nash equilibrium and the pair $(\pi^A, \vec{\theta})$ be jointly a best response to $\pi^P$. Then $\pi^P \in \underset{\pi^P \in \Pi^P}{\operatorname{argmin}}\{ \underset{\pi^A, \vec{\theta} \in \Pi^A \times \Theta^T}{\operatorname{argmax}} \{\text{REGRET}^{\vec{\theta}}(\pi^P, \pi^A)\}\}.$*

*Proof.* Let $(\pi^P, \pi^A, \vec{\theta})$ be in Nash equilibria and the pair $(\pi^A, \vec{\theta})$ is jointly a best response to $\pi^P$. Then we can consider $(\pi^A, \vec{\theta})$ as one player with policy which we will write as $\pi^{A+\vec{\theta}} \in \Pi \times \Theta^T$. Then $\pi^{A+\vec{\theta}}$ is in a zero sum game with $\pi^P$, and the condition that the pair $(\pi^A, \vec{\theta})$ is jointly a best response to $\pi^P$ is equivalent to saying that $\pi^{A+\vec{\theta}}$ is a best response to $\pi^P$. Thus $\pi^P$ and $\pi^{A+\vec{\theta}}$ form a Nash equilibria of a zero sum game, and the minimax theorem applies. Since the reward in this game is defined by REGRET we have:

$$\pi^P \in \underset{\pi^P \in \Pi^P}{\operatorname{argmin}}\{ \underset{\pi^{A+\vec{\theta}} \in \Pi \times \Theta^T}{\operatorname{argmax}} \{\text{REGRET}^{\vec{\theta}}(\pi^P, \pi^A)\}\}$$

By the definition of $\pi^{A+\vec{\theta}}$ this proves the protagonist learns the minimax regret policy. $\square$

The proof gives us reason to believe that the iterative PAIRED training process in Algorithm 1 can produce minimax regret policies. In Appendix D we show that if the agents are in Nash equilibrium, without the coordination assumption, the protagonist will perform at least as well as the antagonist in every parameterization, and Appendix E.1 provides empirical results for alternative methods for approximating regret which break the coordination assumption. In the following sections, we will show that empirically, policies trained with Algorithm 1 exhibit good performance and transfer, both in generating emergent complexity and in training robust policies.

## 5 Experiments

The experiments in this section focus on assessing whether training with PAIRED can increase the complexity of agents' learned behavior, and whether PAIRED agents can achieve better or more

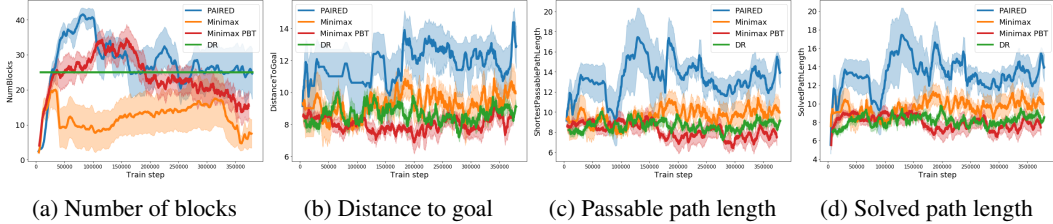

| (a) Number of blocks | (b) Distance to goal | (c) Passable path length | (d) Solved path length |

Figure 2: Statistics of generated environments in terms of the number of blocks (a), distance from the start position to the goal (b), and the shortest path length between the start and the goal, which is zero if there is no possible path (c). The final plot shows agent learning in terms of the shortest path length of a maze successfully solved by the agents. Each plot is measured over five random seeds; error bars are a 95% CI. Domain randomization (DR) cannot tailor environment design to the agent's progress, so metrics remain fixed or vary randomly. Minimax training (even with populations of adversaries and agents) has no incentive to improve agent performance, so the length of mazes that agents are able to solve remains similar to DR (d). In contrast, PAIRED is the only method that continues to increase the passable path length to create more challenging mazes (c), producing agents that solve more complex mazes than the other techniques (d).

robust performance when transferred to novel environments. To study these questions, we first focus on the navigation tasks shown in Figure 1. Section 5.2 presents results in continuous domains.

## 5.1 Partially Observable Navigation Tasks

Here we investigate navigation tasks (based on [9]), in which an agent must explore to find a goal (green square in Figure 1) while navigating around obstacles. The environments are partially observable; the agent's field of view is shown as a blue shaded area in the figure. To deal with the partial observability, we parameterize the protagonist and antagonist's policies using recurrent neural networks (RNNs). All agents are trained with PPO [35]. Further details about network architecture and hyperparameters are given in Appendix F.

We train adversaries that learn to build these environments by choosing the location of the obstacles, the goal, and the starting location of the agent. The adversary's observations consist of a fully observed view of the environment state, the current timestep $t$, and a random vector $z \sim \mathcal{N}(0, I), z \in \mathbb{R}^D$ sampled for each episode. At each timestep, the adversary outputs the location where the next object will be placed; at timestep 0 it places the agent, 1 the goal, and every step afterwards an obstacle. Videos of environments being constructed and transfer performance are available at `https://www.youtube.com/channel/UCI6dkF8eNrCz6XiBJlV9fmw/videos`.

### 5.1.1 Comparison to Prior Methods

To compare to prior work that uses pure minimax training [28, 43, 25] (rather than minimax *regret*), we use the same parameterization of the environment adversary and protagonist, but simply remove the antagonist agent. The adversary's reward is $R(\Lambda) = -\mathbb{E}_{\tau^P}[U(\tau^P)]$. While a direct comparison to POET [45] is challenging since many elements of the POET algorithm are specific to a 2D walker, the main algorithmic distinction between POET and minimax environment design is maintaining a population of environment adversaries and agents that are periodically swapped with each other. Therefore, we also employ a Population Based Training (PBT) minimax technique in which the agent and adversary are sampled from respective populations for each episode. This baseline is our closest approximation of POET [45]. To apply domain randomization, we simply sample the $(x, y)$ positions of the agent, goal, and blocks uniformly at random. We sweep shared hyperparameters for all methods equally. Parameters for the emergent complexity task are selected to maximize the solved path length, and parameters for the transfer task are selected using a set of validation environments. Details are given in the appendix.

### 5.1.2 Emergent Complexity

Prior work [45, 46] focused on demonstrating emergent complexity as the primary goal, arguing that automatically learning complex behaviors is key to improving the sophistication of AI agents. Here,

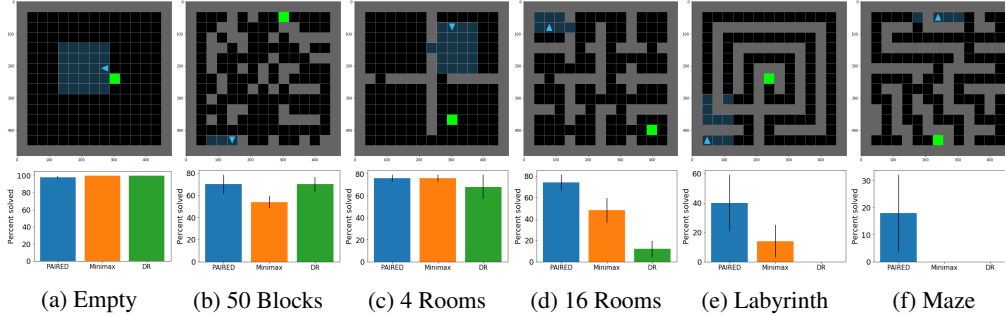

| (a) Empty | (b) 50 Blocks | (c) 4 Rooms | (d) 16 Rooms | (e) Labyrinth | (f) Maze |

Figure 3: Percent successful trials in environments used to test zero-shot transfer, out of 10 trials each for 5 random seeds. The first two (a, b) simply test out-of-distribution generalization to a setting of the number of blocks parameter. Four Rooms (c) tests within-distribution generalization to a specific configuration that is unlikely to be generated through random sampling. The 16 Rooms (d), Labyrinth environment (e) and Maze (f) environments were designed by a human to be challenging navigation tasks. The bar charts show the zero-shot transfer performance of models trained with domain randomization (DR), minimax, or PAIRED in each of the environments. Error bars show a 95% confidence interval. As task difficulty increases, only PAIRED retains its ability to generalize to the transfer tasks.

we track the complexity of the generated environments and learned behaviors throughout training. Figure 2 shows the number of blocks (a), distance to the goal (b), and the length of the shortest path to the goal (c) in the generated environments. The solved path length (d) tracks the shortest path length of a maze that the agent has completed successfully, and can be considered a measure of the complexity of the agent's learned behavior.

Domain randomization (DR) simply maintains environment parameters within a fixed range, and cannot continue to propose increasingly difficult tasks. Techniques based on minimax training are purely motivated to decrease the protagonist's score, and as such do not enable the agent to continue learning; both minimax and PBT obtain similar performance to DR. In contrast, PAIRED creates a curriculum of environments that begin with shorter paths and fewer blocks, but gradually increase in complexity based on both agents' current level of performance. As shown in Figure 2 (d), this allows PAIRED protagonists to learn to solve more complex environments than the other two techniques.

### 5.1.3 Zero-Shot Transfer

In order for RL algorithms to be useful in the real world, they will need to generalize to novel environment conditions that their developer was not able to foresee. Here, we test the zero-shot transfer performance on a series of novel navigation tasks shown in Figure 3. The first two transfer tasks simply use a parameter setting for the number of blocks that is out of the distribution (OOD) of the training environments experienced by the agents (analogous to the evaluation of [28]). We expect these transfer scenarios to be easy for all methods. We also include a more structured but still very simple Four Rooms environment, where the number of blocks is within distribution, but in an unlikely (though simple) configuration. As seen in the figure, this can usually be completed with nearly straight-line paths to the goal. To evaluate difficult transfer settings, we include the 16 rooms, Labyrinth, and Maze environments, which require traversing a much longer and more challenging path to the goal. This presents a much more challenging transfer scenario, since the agent must learn meaningful navigation skills to succeed in these tasks.

Figure 3 shows zero-shot transfer performance. As expected, all methods transfer well to the first two environments, although the minimax adversary is significantly worse in 50 Blocks. Varying the number of blocks may be a relatively easy transfer task, since partial observability has been shown to improve generalization performance in grid-worlds [47]. As the environments increase in difficulty, so does the performance gap between PAIRED and the prior methods. In 16 Rooms, Labyrinth, and Maze, PAIRED shows a large advantage over the other two methods. In the Labyrinth (Maze) environment, PAIRED agents are able to solve the task in 40% (18%) of trials. In contrast, minimax agents solve 10% (0.0%) of trials, and DR agents solve 0.0%. The performance of PAIRED on these tasks can be explained by the complexity of the generated environments; as shown in Figure 1c, the

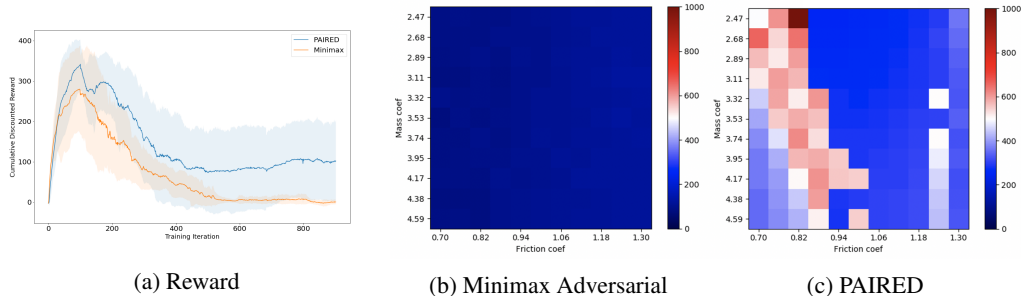

|  |  |  |
|:---:|:---:|:---:|
| (a) Reward | (b) Minimax Adversarial | (c) PAIRED |

Figure 4: Training curves for PAIRED and minimax adversaries on the MuJoCo hopper, where adversary strength is scaled up over the first 300 iterations (a). While both adversaries reduce the agent's reward by making the task more difficult, the minimax adversary is incentivized to drive the agent's reward as low as possible. In contrast, the PAIRED adversary must maintain feasible parameters. When varying force and mass are applied at test time, the minimax policy performs poorly in cumulative reward (b), while PAIRED is more robust (c).

field of view of the agent (shaded blue) looks similar to what it might encounter in a maze, although the idea of a maze was never explicitly specified to the agent, and this configuration blocks is very unlikely to be generated randomly. These results suggest that PAIRED training may be better able to prepare agents for challenging, unknown test settings.

## 5.2   Continuous Control Tasks

To compare more closely with prior work on minimax adversarial RL [28, 43], we construct an additional experiment in a modified version of the MuJoCo hopper domain [42]. Here, the adversary outputs additional torques to be applied to each joint at each time step, $\vec{\theta}$. We benchmark PAIRED against unconstrained minimax training. To make minimax training feasible, the torque the adversary can apply is limited to be a proportion of the agent's strength, and is scaled from 0.1 to 1.0 over the course of 300 iterations. After this point, we continue training with no additional mechanisms for constraining the adversary. We pre-train the agents for 100 iterations, then test transfer performance to a set of mass and friction coefficients. All agents are trained with PPO [35] and feed-forward policies.

Figure 4a shows the rewards throughout training. We observe that after the constraints on the adversaries are removed after 300 iterations, the full-strength minimax adversary has the ability to make the task unsolvable, so the agent's reward is driven to zero for the rest of training. In contrast, PAIRED is able to automatically adjust the difficulty of the adversary to ensure the task remains solvable. As expected, Figure 4 shows that when constraints on the adversary are not carefully tuned, policies trained with minimax fail to learn and generalize to unseen environment parameters. In contrast, PAIRED agents are more robust to unseen environment parameters, even without careful tuning of the forces that the adversary is able to apply.

## 6   Conclusions

We develop the framework of Unsupervised Environment Design (UED), and show its relevance to a range of RL tasks, from learning increasingly complex behavior or more robust policies, to improving generalization to novel environments. In environments like games, for which the designer has an accurate model of the test environment, or can easily enumerate all of the important cases, using UED may be unnecessary. However, UED could provide an approach to building AI systems in many of the ambitious uses of AI in real-world settings which are difficult to accurately model.

We have shown that tools from the decision theory literature can be used to characterize existing approaches to environment design, and motivate a novel approach based on minimax regret. Our algorithm, which we call Protagonist Antagonist Induced Regret Environment Design (PAIRED), avoids common failure modes of existing UED approaches, and is able to generate a curriculum of increasingly complex environments. Our results demonstrate that PAIRED agents learn more complex behaviors, and achieve higher zero-shot transfer performance in challenging, novel environments.

## Broader Impact

Unsupervised environment design is a technique with a wide range of applications, including unsupervised RL, transfer learning, and Robust RL. Each of these applications has the chance of dramatically improving the viability of real-world AI systems. Real-world AI systems could have a variety of positive impacts, reducing the possibility for costly human error, increasing efficiency, and doing tasks that are dangerous or difficult for humans. However, there are a number of possible negative impacts: increasing unemployment by automating jobs [11] and improving the capabilities of automated weapons. These positives and negatives are potentially exacerbated by the emergent complexity we described in Section 5.1.2, which could lead to improved efficiency and generality allowing the robotics systems to be applied more broadly.

However, if we are to receive any of the benefits of AI powered systems being deployed into real world settings, it is critical that we know how to make these systems robust and that they know how to make good decisions in uncertain environments. In Section 3 we discussed the deep connections between UED and *decisions under ignorance*, which shows that reasonable techniques for making decisions in uncertain settings correspond to techniques for UED, showing that the study of UED techniques can be thought of as another angle of attack at the problem of understanding how to make robust systems. Moreover, we showed in Section 5.2 that these connections are not just theoretical, but can lead to more effective ways of building robust systems. Continued work in this area can help ensure that the predominant impact of our AI systems is the impact we intended.

Moreover, many of the risks of AI systems come from the system acting unexpectedly in a situation the designer had not considered. Approaches for Unsupervised Environment Design could work towards a solution to this problem, by automatically generating interesting and challenging environments, hopefully detecting troublesome cases before they appear in deployment settings.

## Acknowledgments and Disclosure of Funding

We would like to thank Michael Chang, Marvin Zhang, Dale Schuurmans, Aleksandra Faust, Chase Kew, Jie Tan, Dennis Lee, Kelvin Xu, Abhishek Gupta, Adam Gleave, Rohin Shah, Daniel Filan, Lawrence Chan, Sam Toyer, Tyler Westenbroek, Igor Mordatch, Shane Gu, DJ Strouse, and Max Kleiman-Weiner for discussions that contributed to this work. We are grateful for funding of this work as a gift from the Berkeley Existential Risk Intuitive. We are also grateful to Google Research for funding computation expenses associated with this work.

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
