[Supplementary Material]

# A  Generality of UED

In Section 3 we defined UPODMPS and UED and showed a selection of natural approaches to UED and corrisponding decision rules. However, the connection between UED and decisions under ignorance is much broader than these few decision rules. In fact, they can be seen as nearly identical problems.

To see the connection to decisions under ignorance, it is important to notice that any decision rule can be thought of as an ordering over policies, ranking the policies that it chooses higher than other policies. We will want to define a condition on this ordering, to do this we will first define what it means for one policy to *totally dominate* another.

**Definition 3.** *A policy, $\pi_A$, is totally dominated by some policy, $\pi_B$ if for every pair of parameterizations $\vec{\theta}_A, \vec{\theta}_B$ $U^{\mathcal{M}_{\vec{\theta}_A}}(\pi_A) < U^{\mathcal{M}_{\vec{\theta}_B}}(\pi_B)$.*

Thus if $\pi_A$ totally dominates $\pi_B$, it is reasonable to assume that $\pi_A$ is better, since the best outcome we could hope for from policy $\pi_B$ is still worse than the worst outcome we fear from policy $\pi_A$. Thus we would hope that our decision rule would not prefer $\pi_B$ to $\pi_A$. If a decision rule respects this property we will say that it *respects total domination* or formally:

**Definition 4.** *We will say that an ordering $\prec$ respects total domination iff $\pi_A \prec \pi_B$ whenever $\pi_B$ totally dominates $\pi_A$.*

This is a very weak condition, but it is already enough to allow us to provide a characterization of all such orderings over policies in terms of policy-conditioned distributions over parameters $\vec{\theta}$, which we will notate $\Lambda$ as in Section 3. Specifically, any ordering which respects total domination can be written as maximizing the expected value with respect to a *policy-conditioned value function*, thus every reasonable way of ranking policies can be described in the UED framework. For example, this implies that there is an environment policy which represents the strategy of minimax regret. We explicitly construct one such environment policy in Appendix B.

To make this formal, the *policy-conditioned value function*, $V^{\mathcal{M}_\Lambda}(\pi)$, is defined to be the expected value a policy will receive in the policy-conditioned distribution of environments $\mathcal{M}_{\Lambda(\pi)}$, or formally:

$$V^{\mathcal{M}_\Lambda}(\pi) = \mathbb{E}_{\vec{\theta} \sim \Lambda(\pi)}[U^{\vec{\theta}}(\pi)] \tag{2}$$

.

The policy-conditioned value function is like the normal value function, but computed over the distribution of environments defined by the UPOMDP and environment policy. This of course implies an ordering over policies, defined in the natural way.

**Definition 5.** *We will say that $\Lambda$ prefers $\pi_B$ to $\pi_A$ notated $\pi_A \prec^\Lambda \pi_B$ if $V^{\mathcal{M}_\Lambda}(\pi_A) < V^{\mathcal{M}_\Lambda}(\pi_B)$*

Finally, this allows us to state the main theorem of this Section.

**Theorem 6.** *Given an order over deterministic policies in a finite UPOMDP, $\prec$, there exits an environment policy $\Lambda : \Pi \to \mathbf{\Delta}(\Theta^T)$ such that $\prec$ is equivalent to $\prec^\Lambda$ iff it respects total domination and it ranks policies with an equal and a deterministic outcome as equal.*

*Proof.* Suppose you have some order of policies in a finite UPOMDP, $\prec$, which respects total domination and ranks policies equal if they have an equal and deterministic outcome. Our goal is to construct a function $\Lambda$ such that $\pi_A \prec^\Lambda \pi_B$ iff $\pi_A \prec \pi_B$.

When constructing $\Lambda$ notice that we can chose $\Lambda(\pi)$ independently for each policy $\pi$ and that we can choose the resulting value for $V^{\mathcal{M}_\Lambda}(\pi)$ to lie anywhere within the range $[\min_{\vec{\theta} \in \Theta^T}\{U^{\vec{\theta}}(\pi)\}, \max_{\vec{\theta} \in \Theta^T}\{U^{\vec{\theta}}(\pi)\}]$. Since the number of deterministic policies in a finite POMDP is finite, we can build $\Lambda$ inductively by taking the lowest ranked policy $\pi$ in terms of $\prec$ for which $\Lambda$ has not yet been defined and choosing the value for $V^{\mathcal{M}_\Lambda}(\pi)$ appropriately.

For the lowest ranked policy $\pi_B$ for which $\Lambda(\pi_B)$ has not yet been defined we set $\Lambda(\pi_B)$ such that $V^{\mathcal{M}_\Lambda}(\pi_B)$ greater than $\Lambda(\pi_A)$ for all $\pi_A \prec \pi_B$ and is lower than the minimum possible value of

any $\pi_C$ such that $\pi_B \prec \pi_C$, if such a setting is possible. That is, we choose $\Lambda(\pi_B)$ to satisfy for all $\pi_A \prec \pi_B \prec \pi_C$:

$$\Lambda(\pi_A) < V^{\mathcal{M}_\Lambda}(\pi_B) < \max_{\vec{\theta} \in \Theta^T}\{U^{\vec{\theta}}(\pi_C)\} \tag{3}$$

Intuitively this ensures that $V^{\mathcal{M}_\Lambda}(\pi_B)$ is high enough to be above all $\pi_A$ lower than $\pi_B$ and low enough such that all future $\pi_C$ can still be assigned an appropriate value.

Finally, we will show that it is possible to set $\Lambda(\pi_B)$ to satisfy these conditions in Equation 3. By our inductive hypothesis, we know that $\Lambda(\pi_A) < \max_{\vec{\theta} \in \Theta^T}\{U^{\vec{\theta}}(\pi_B)\}$ and $\Lambda(\pi_A) < \max_{\vec{\theta} \in \Theta^T}\{U^{\vec{\theta}}(\pi_C)\}$. Since $\prec$ respects total domination, we know that $\min_{\vec{\theta} \in \Theta^T}\{U^{\vec{\theta}}(\pi_B)\} \le \max_{\vec{\theta} \in \Theta^T}\{U^{\vec{\theta}}(\pi_C)\}$ for all $\pi_B \le \pi_C$. Since there are a finite number of $\pi_C$ we can set $V^{\mathcal{M}_\Lambda}(\pi_B)$ to be the average of the smallest value for $\max_{\vec{\theta} \in \Theta^T}\{U^{\vec{\theta}}(\pi_C)\}$ and the largest value for $\Lambda(\pi_A)$ for any $\pi_C$ and $\pi_A$ satisfying $\pi_A \prec \pi_B \prec \pi_C$.

The other direction, can be checked directly. If $\pi_A$ is totally dominated by $\pi_B$, then:

$$V^{\mathcal{M}_\Lambda}(\pi_A) = \mathbb{E}_{\vec{\theta} \sim \Lambda(\pi_A)}[U^{\vec{\theta}}(\pi_A)] < \mathbb{E}_{\vec{\theta} \sim \Lambda(\pi_B)}[U^{\vec{\theta}}(\pi_B)] = V^{\mathcal{M}_\Lambda}(\pi_B)$$

Thus if $\prec$ can be represented with an appropriate choice of $\Lambda$ then it respects total domination and it ranks policies with an equal and a deterministic outcome equal. $\square$

Thus, the set of decision rules which respect total domination are exactly those which can be written by some concrete environment policy, and thus any approach to making decisions under uncertainty which has this property can be thought of as a problem of unsupervised environment design and vice-versa. To the best of our knowledge there is no seriously considered approach to making decisions under uncertainty which does not satisfy this property.

## B  Defining an Environment Policy Corresponding to Minimax Regret

In this section we will be deriving a function $\Lambda(\pi)$ which corresponds to regret minimization. We will be assuming a finite MDP with a bounded reward function. Explicitly, we will be deriving a $\Lambda$ such that minimax regret is the decision rule which maximizes the corresponding *policy-conditioned value function* defined in Appendix A:

$$V^{\mathcal{M}_\Lambda}(\pi) = \mathbb{E}_{\vec{\theta} \sim \Lambda(\pi)}[U^{\vec{\theta}}(\pi)]$$

In this context, we will define regret to be:

$$\text{REGRET}(\pi, \vec{\theta}) = \max_{\pi^B \in \Pi}\{U^{\mathcal{M}_{\vec{\theta}}}(\pi^B) - U^{\mathcal{M}_{\vec{\theta}}}(\pi)\}$$

Thus the set of MINIMAXREGRET strategies can be defined naturally as the ones which achive the minimum worst case regret:

$$\text{MINIMAXREGRET} = \operatorname*{argmin}_{\pi \in \Pi}\{\max_{\vec{\theta} \in \Theta^T}\{\text{REGRET}(\pi, \vec{\theta})\}\}$$

We want to find a function $\Lambda$ for which the set of optimal policies which maximize value with respect to $\Lambda$ is the same as the set of MINIMAXREGRET strategies. That is, we want to find $\Lambda$ such that:

$$\text{MINIMAXREGRET} = \operatorname*{argmax}_{\pi \in \Pi}\{V^{\mathcal{M}_\Lambda}(\pi)\}$$

To do this it is useful to first introduce the concept of *weak total domination*, which is the natural weakening of the concept of total domination introduced in Appendix A. A policy is said to be weakly totally dominated by a policy $\pi_B$ if the maximum outcome that can be achieved by $\pi_A$ is equal to the minimum outcome that can be achieved by $\pi_B$, or formally:

**Definition 7.** *A policy, $\pi_A$, is* weakly totally dominated *by some policy, $\pi_B$ if for every pair of parameterizations $\vec{\theta}_A, \vec{\theta}_B$ $U^{\mathcal{M}_{\vec{\theta}_A}}(\pi_A) \leq U^{\mathcal{M}_{\vec{\theta}_B}}(\pi_B)$.*

The concept of weak total domination is only needed here to clarify what happens in the special case that there are policies that are weakly totally dominated but not totally dominated. These policies may or may not be minimax regret optimal policies, and thus have to be treated with more care. To get a general sense for how the proof works you may assume that there are no policies which are weakly totally dominated but not totally dominated and skip the sections of the proof marked as a special case: the second paragraph of the proof for Lemma 8 and the second paragraph of the proof for Theorem 9.

We can use this concept to help define a normalization function $D : \Pi \to \mathbf{\Delta}(\Theta^T)$ which has the property that if there are two policies $\pi_A, \pi_B$ such that neither is totally dominated by the other, then they evaluate the same value on the distribution of environments. We will use $D$ as a basis for creating $\Lambda$ by shifting probability mass towards or away from this distribution in the right proportion. In general there are many such normalization functions, but we will simply show that one of these exists.

**Lemma 8.** *There exists a function $D : \Pi \to \mathbf{\Delta}(\Theta^T)$ such that $\Lambda(\pi)$ has support at least $s > 0$ on the highest-valued regret-maximizing parameterization $\overline{\theta}_\pi$ for all $\pi$, that for all $\pi_A, \pi_B$ such that neither is weakly totally dominated by any other policy, $V^{\mathcal{M}_D}(\pi_A) = V^{\mathcal{M}_D}(\pi_B)$ and $V^{\mathcal{M}_D}(\pi_A) > V^{\mathcal{M}_D}(\pi_B)$ when $\pi_B$ is totally dominated and $\pi_A$ is not. If the policy $\pi$ is weakly dominated but not totally dominated we choose $D$ to put full support on the highest-valued regret-maximizing parameterization, $D(\pi) = \overline{\theta}_\pi$.*

*Proof.* We will first define $D$ for the set of policies which are not totally dominated, which we will call $X$. Note that by the definition of not being totally dominated, there is a constant $C$ which is between the worst-case and best-case values of all of the policies in $X$.

**Special Case:** If there is only one choice for $C$ and if that does have support over $\overline{\theta}_\pi$ for each $\pi$ which is not totally dominated, then we know that there is a weakly totally dominated policy which is not totally dominated. In this case we choose a $C$ which is between the best-case and the worst-case for the not weakly totally dominated policies. Thus for each $\pi$ which is not weakly dominated we can find a distribution $\vec{\theta}$ such that $U^{\vec{\theta}}(\pi) = C$. If C is not the maximum or minimum achievable value for $\pi$ then we will chose $D(\pi)$ to have expected value $C$ and and have support over all $\Theta^T$.

Otherwise, we can choose $D(\pi) = \overline{\theta}_\pi$. For other $\pi \notin X$, we can let $D(\pi) = U(\Theta^T)$, which achieves value less than $C$ since all outcomes for $\pi$ have utility less than $C$. Thus, by construction $D$ satisfies the desired conditions. $\square$

Given such a function $D$, we will construct a $\Lambda$ which works by shifting probability towards or away from the environment parameters that maximize the regret for the given agent, in the right proportions to achieve the desired result. We claim that the following definition works:

$$\Lambda(\pi) = \{\overline{\theta}_\pi : \frac{c_\pi}{v_\pi}, \tilde{D}_\pi : \text{otherwise}\}$$

Where the bracket notation defines a mixture distribution $\{x : P, y : Q\}(\theta) = xP(\theta) + yQ(\theta)$ and where $\tilde{D}_\pi = D(\pi)$ is a baseline distribution satisfying the conditions of Lemma 8, $\overline{\theta}_\pi$ is the trajectory which maximizes regret of $\pi$, and $v_\pi$ is the value above the baseline distribution that $\pi$ achieves on that trajectory, and $c_\pi$ is the negative of the worst-case regret of $\pi$, normalized so that $\frac{c_\pi}{v_\pi}$ is between $-s$ and 1. If $\frac{c_\pi}{v_\pi}$ is negative then we will interpret probability mass of $\frac{c_\pi}{v_\pi}$ on $\overline{\theta}_\pi$ in $\tilde{D}_\pi$ being redistributed proportionally in $\tilde{D}_\pi$ as is implied by the equation. Note that if $v_\pi = 0$ then $\tilde{D}_\pi = \overline{\theta}_\pi$ by our construction, so $\frac{c_\pi}{v_\pi}$ can be interpreted to be anything in $[0, 1]$ without effecting the distribution $\Lambda(\pi)$.

**Theorem 9.** *For $\Lambda$ as defined above:*

$$\operatorname*{argmax}_{\pi \in \Pi}\{\underset{\vec{\theta} \sim \Lambda(\pi)}{\mathbb{E}}[U^{\vec{\theta}}(\pi)]\} = \textsc{MinimaxRegret}$$

*Proof.* By the construction of $D(\pi)$ by Lemma 8, we will let $C = \mathbb{E}_{\vec{\theta} \sim \tilde{D}_\pi}[U^{\vec{\theta}}(\pi)]$, which is a constant independent of $\pi$ by construction for all of the policies which are not weakly totally dominated. If there are no policies which are weakly totally dominated but not totally dominated the next paragraph can be skipped, otherwise it handles the special case by showing that the conditions we ensured for the non-totally dominated policies hold for these weakly totally dominated policies if and only if they are minimax regret optimal policies.

**Special Case:** For the $\pi$ which are weakly totally dominated, but not totally dominated, we will show that $\mathbb{E}_{\vec{\theta} \sim \tilde{D}_\pi}[U^{\vec{\theta}}(\pi)] = C$ if $\pi \in$ MINIMAXREGRET and $\mathbb{E}_{\vec{\theta} \sim \tilde{D}_\pi}[U^{\vec{\theta}}(\pi)] < C$ if $\pi \notin$ MINIMAXREGRET. Let $M$ be the maximum value achieved by any policy. Then if $\pi \in$ MINIMAXREGRET and the maximum value it achieves is $C'$ then note that any policy that weakly dominates $\pi$, say $\pi'$, can achieve no more than $M - C'$ regret. Thus if $\pi$ is a minimax optimal strategy, it must achieve exactly $M - C'$ regret and both $\pi$ and $\pi'$ must have a regret maximizing outcome at the same value $M - C'$, which is the maximum value outcome for $\pi$ and the minimum value outcome for $\pi'$. Thus $C = C'$ since both $\pi$ and $\pi'$ must have support on the an outcome of value $C'$ and $\mathbb{E}_{\vec{\theta} \sim \tilde{D}_\pi}[U^{\vec{\theta}}(\pi)] = C$.

Otherwise $\pi \notin$ MINIMAXREGRET so it achieves more than $M - C'$ regret and $\mathbb{E}_{\vec{\theta} \sim \tilde{D}_\pi}[U^{\vec{\theta}}(\pi)] < C$.

With this case out of the way, we can use the definition of $\Lambda$ and simplify to show the desired result.

$$\operatorname*{argmax}_{\pi \in \Pi}\{\mathbb{E}_{\vec{\theta} \sim \Lambda(\pi)}[U^{\vec{\theta}}(\pi)]\} = \operatorname*{argmax}_{\pi \in \Pi}\{\frac{U^{\bar{\theta}_\pi}(\pi)c_\pi}{v_\pi} + C(1 - \frac{c_\pi}{v_\pi})\} \tag{4}$$

$$= \operatorname*{argmax}_{\pi \in \Pi}\{\frac{v_\pi c_\pi + C c_\pi}{v_\pi} + (\frac{C v_\pi - C c_\pi}{v_\pi})\} \tag{5}$$

$$= \operatorname*{argmax}_{\pi \in \Pi}\{\frac{v_\pi c_\pi + C v_\pi}{v_\pi}\} \tag{6}$$

$$= \operatorname*{argmax}_{\pi \in \Pi}\{c_\pi + C\} \tag{7}$$

$$= \operatorname*{argmax}_{\pi \in \Pi}\{c_\pi\} \tag{8}$$

$$= \operatorname*{argmax}_{\pi \in \Pi}\{\min_{\vec{\theta} \in \Theta^T}\{-\text{REGRET}(\pi, \vec{\theta})\}\} \tag{9}$$

$$= \operatorname*{argmin}_{\pi \in \Pi}\{\max_{\vec{\theta} \in \Theta^T}\{\text{REGRET}(\pi, \vec{\theta})\}\} \tag{10}$$

$\square$

# C Minimax Regret Always Succeeds when There is a Clear Notion of Success

In this section we will show that when there is a sufficiently strong notion of success and failure, and there is a policy which can ensure success, minimax regret will choose a successful strategy. Moreover, we will show that neither pure randomization, nor maximin has this property.

**Theorem 1.** *Suppose that all achievable rewards fall into one of two class of outcomes labeled **SUCCESS** giving rewards in $[\mathbf{S}_{min}, \mathbf{S}_{max}]$ and **FAILURE** giving rewards in $[\mathbf{F}_{min}, \mathbf{F}_{max}]$, such that $\mathbf{F}_{min} \leq \mathbf{F}_{max} < \mathbf{S}_{min} \leq \mathbf{S}_{max}$. In addition assume that the range of possible rewards in either class is smaller than the difference between the classes so we have $\mathbf{S}_{max} - \mathbf{S}_{min} < \mathbf{S}_{min} - \mathbf{F}_{max}$ and $\mathbf{F}_{max} - \mathbf{F}_{min} < \mathbf{S}_{min} - \mathbf{F}_{max}$. Further suppose that there is a policy $\pi$ which succeeds on any $\vec{\theta}$ whenever success is possible. Then minimax regret will choose a policy which has that property.*

*Further suppose that there is a policy $\pi$ which succeeds on any $\vec{\theta}$ whenever any policy succeeds on $\vec{\theta}$. Then minimax regret will choose a policy which has that property.*

*Proof.* Let $C = \mathbf{S}_{min} - \mathbf{F}_{max}$ and let $\pi^*$ be one of the policies that succeeds on any $\vec{\theta}$ whenever success is possible. By assumption $\mathbf{S}_{max} - \mathbf{S}_{min} < C$ and $\mathbf{F}_{max} - \mathbf{F}_{min} < C$. Since $\pi^*$ succeeds

whenever success is possible we have:

$$\max_{\vec{\theta} \in \Theta^T} \{ \text{REGRET}(\pi^*, \vec{\theta}) \} < C$$

Thus any strategy which achieves minimax regret must have regret less than $C$, since it must do at least as well as $\pi^*$. Suppose $\pi \in$ MINIMAXREGRET, then if $\pi$ does not solve some solvable $\vec{\theta}$ then REGRET$(\pi^*, \vec{\theta}) > C$. Thus every minimax regret policy succeeds whenever that is possible. $\qquad\square$

Though minimax regret has this property, the other decision rules considered do not. For example, consider the task described by Table 3a, where an entry in position $(i, j)$ shows the payoff $U^{\theta_j}(\pi_i)$ obtained for the policy $\pi_i$ in row $i$, in the environment parameterized by $\theta_j$ in column $j$. If you choose $\pi_B$ you can ensure success whenever any policy can ensure success, but maximin will choose $\pi_A$, as it only cares about the worst case outcome.

|         | $\theta_1$ | $\theta_2$ |
|---------|-----------|-----------|
| $\pi_A$ | 0         | 0         |
| $\pi_B$ | 100       | -1        |

(a)

|         | $\theta_1$ | $\theta_2$ | $\theta_3$ | $\theta_4$ | $\theta_5$ |
|---------|-----------|-----------|-----------|-----------|-----------|
| $\pi_A$ | 75        | 75        | 75        | 75        | 75        |
| $\pi_B$ | 0         | 100       | 100       | 100       | 100       |
| $\pi_C$ | 100       | 0         | 100       | 100       | 100       |
| $\pi_D$ | 100       | 100       | 0         | 100       | 100       |
| $\pi_E$ | 100       | 100       | 100       | 0         | 100       |
| $\pi_F$ | 100       | 100       | 100       | 100       | 0         |

(b)

Table 3: In these setting, **SUCCESS** is scoring in the range $[75, 100]$ and **FAILURE** is scoring in the range $[-1, 0]$

Choosing the policy that maximizes the expected return fails in the game described by Table 3b. Under a uniform distribution all of $\pi_B, \pi_C, \pi_D, \pi_E, \pi_F$ give an expected value of 80, while $\pi_A$ gives an expected value of 75. Here maximizing expected value on a uniform distribution will choose one of $\pi_B, \pi_C, \pi_D, \pi_E, \pi_F$ while $\pi_A$ guarantees success. Moreover, this holds for every possible distribution over parameters, since $\pi_A$ will always give an expected value of 75 and on average $\pi_B, \pi_C, \pi_D, \pi_E, \pi_F$ will give an expected value of 80, so at least one of them gives above 80.

## D  Nash solutions to PAIRED

In this section we show how PAIRED works when coordination is not achieved. The following proof shows that if the antagonist, protagonist, and adversary find a Nash Equilibrium, then the protagonist performs better than or equal to the antagonist in every parameterization.

**Theorem 10.** *Let $(\pi^P, \pi^A, \vec{\theta})$ be in Nash equilibria. Then for all $\vec{\theta'}$: $U^{\vec{\theta'}}(\pi_P) \geq U^{\vec{\theta'}}(\pi_A)$.*

*Proof.* Let $(\pi^P, \pi^A, \vec{\theta})$ be in Nash equilibria. Then regardless of $\pi^A, \vec{\theta}$, the protagonist always has the choice to play $\pi^P = \pi^A$, so REGRET$(\pi^P, \pi^A, \vec{\theta}) \leq 0$. Since, the adversary chooses $\vec{\theta}$ over any other $\vec{\theta'}$ we know that REGRET$(\pi^P, \pi^A, \vec{\theta'}) \leq 0$ must hold for all $\vec{\theta'}$. Thus by the definition of Regret $U^{\vec{\theta'}}(\pi_P) \geq U^{\vec{\theta'}}(\pi_A)$. $\qquad\square$

This shows that with a capable antagonist the protagonist would learn the minimax regret policy, even without the adversary and the antagonist coordinating, and suggests that methods which strengthen the antagonist could serve to improve the protagonist.

## E  Additional experiments

### E.1  Alternative methods for computing regret

Given the arguments in Appendix D, we experimented with additional methods for approximating the regret, which attempt to improve the antagonist. We hypothesized that using a fixed agent as

| (a) Number of blocks | (b) Distance to goal | (c) Passable path length | (d) Solved path length |

Figure 5: Comparison of alternative methods for computing the regret in terms of the statistics of generated environments (a-c), and the length of mazes that agents are able to solve (d). Neither the combined population or flexible approach (which both break the coordination assumption in Theorem 2) show improved complexity. Statistics of generated and solved environments measured over five random seeds; error bars are a 95% CI.

the antagonist could potentially lead to optimization problems in practice; if the antagonist became stuck in a local minimum and stopped learning, it could limit the complexity of the environments the adversary could generate. Therefore, we developed an alternative approach using population-based training (PBT), where for each environment designed by the adversary, each agent in the population collects a single trajectory in the environment. Then, the regret is computed using the difference between the maximum performing agent in the population, and the mean of the population. Assume there is a population of $K$ agents, and $i$ and $j$ index agents within the population. Then:

$$\text{REGRET}_{pop} = \max_i U(\tau^i) - \frac{1}{K} \sum_{j=1}^{K} [U(\tau^j)] \qquad (11)$$

We also used a population of adversaries, where for each episode, we randomly select an adversary to generate the environment, then test all agents within that environment. We call this approach *PAIRED Combined PBT*.

Since using PBT can be expensive, we also investigated using a similar approach in the case of a population of $K = 2$ agents, and a single adversary. This is analogous to a version of PAIRED where there is no fixed antagonist, but the antagonist is flexibly chosen to be the currently best-performing agent. We call this approach *Flexible PAIRED*.

Figure 5 plots the complexity results of these two approaches. We find that they both perform worse than the proposed version of PAIRED. Figure 6 shows the transfer performance of both methods against the original PAIRED. Both methods achieve reasonable transfer performance, retaining the ability to solve complex test tasks like the labyrinth and maze (when domain randomization and minimax training cannot). However, we find that the original method for computing the regret outperforms both approaches. We note that both the combined population and flexible PAIRED approaches do not enable the adversary and antagonist to coordinate (since the antagonist does not remain fixed). Coordination between the antagonist and adversary is an assumption in Theorem 2 of Section 4, which shows that if the agents reach a Nash equilibrium, the protagonist will be playing the minimum regret policy. These empirical results appear to indicate that when coordination between the adversary and antagonist is no longer possible, performance degrades. Future work should investigate whether these results hold across more domains.

### E.2  Should agents themselves optimize regret?

We experimented with whether the protagonist and antagonist should optimize for regret, or for the normal reward supplied by the environment (note that the environment-generating adversary always optimizes for the regret). For both the protagonist and antagonist, the regret is based on the difference between their reward for the current trajectory, and the max reward of the other agent received over several trajectories played in the current environment. Let the current agent be $A$, and the other agent be $O$. Then agent $A$ should receive reward $R^A = U(\tau^A) - \max_{\tau^O} U(\tau^O)$ for episode $\tau^A$. However, this reward is likely to be negative. If given at the end of the trajectory, it would essentially punish the agent for reaching the goal, making learning difficult. Therefore we instead compute a per-timestep

(a) Empty      (b) 50 Blocks      (c) 4 Rooms      (d) 16 Rooms      (e) Labyrinth      (f) Maze

Figure 6: Comparison of transfer performance for alternative methods for computing the regret. While both alternatives still retain the ability to solve complex tasks, achieving superior performance to minimax and domain randomization, their performance is not as good as the proposed method for computing the regret.

(a) Number of blocks      (b) Distance to goal      (c) Passable path length      (d) Solved path length

Figure 7: Comparison of alternative methods for computing the regret in terms of the statistics of generated environments (a-c), and the length of mazes that agents are able to solve. Neither the combined population or flexible approach show improved complexity. Statistics of generated and solved environments measured over five random seeds; error bars are a 95% CI.

penalty of $\frac{\max_{\tau^O} U(\tau^O)}{T}$, where $T$ is the maximum length of an episode. This is subtracted post-hoc from each agent's reward at every step during each episode, after the episode has been collected but before the agent is trained on it. When the agent reaches the goal, it receives the normal Minigrid reward for successfully navigating to the goal, which is $R = 1 - 0.9 * (M/T)$, where $M$ is the timestep on which the agent found the goal.

Figure 7 shows the complexity results for the best performing hyperparameters in which the protagonist and antagonist optimized the regret according to the formula above, or whether they simply learned according to the environment reward. Note that in both cases, the environment-generating adversary optimizes the regret of the protagonist with respect to the antagonist. As is evident in Figure 7, training agents on the environment reward itself, rather than the regret, appears to be more effective for learning complex behavior. We hypothesize this is because the regret is very noisy. The performance of the other agent is stochastic, and variations in the other agent's reward are outside of the agent's control. Further, agents do not receive observations about the other agent, and cannot use them to determine what is causing the reward to vary. However, we note that optimizing for the regret can provide good transfer performance. The transfer plots in Figure 3 were created with an agent that optimized for regret, as we describe below. It is possible that as the other agent converges, the regret provides a more reliable signal indicating when the agent's performance is sub-optimal.

# F    Experiment Details and Hyperparameters

## F.1    Navigation Experiments

### F.1.1    Agent parameterization

The protagonist and antagonist agents for both PAIRED and the baselines received a partially observed, $5 \times 5 \times 3$ view of the environment, as well as an integer ranging from $0 - 3$ representing

the direction the agent is currently facing. The agents use a network architecture consisting of a single convolutional layer which connects to an LSTM, and then to two fully connected layers which connect to the policy outputs. A second network with identical parameters is used to estimate the value function. The agents use convolutional kernels of size 3 with 16 filters to input the view of the environment, an extra fully-connected layer of size 5 to process the direction and input it to the LSTM, an LSTM of size 256, and two fully connected layers of size 32 which connect to either the policy outputs or the value estimate. The best entropy regularization coefficient was found to be 0.0. All agents (including environment adversaries) are trained with PPO with a discount factor of 0.995, a learning rate of 0.0001, and 30 environment workers operating in parallel to collect a batch of episodes, which is used to complete one training update.

### F.1.2 Adversary parameterization

The environments explored in this paper are a $15 \times 15$ tile discrete grid, with a border of walls around the edge. This means there are $13 \times 13 = 169$ free tiles for the adversary to use to place obstacles. We parameterized the adversary by giving it an action space of dimensionality 169, and each discrete action indicates the location of the next object to be placed. It plays a sequence of actions, such that on the first step it places the agent, on the second it places the goal, and for 50 steps afterwards it places a wall (obstacle). If the adversary places an object on top of a previously existing object, its action does nothing; this allows it to place fewer than 50 obstacles. If it tries to place the goal on top of the agent, the goal will be placed randomly.

We also explored an alternative parameterization, in which the adversary had only 4 actions, which corresponded to placing the agent, goal, an obstacle, or nothing. It then took a sequence of 169 steps to place all objects, moving through the grid from top to bottom and left to right. If it chose to place the agent or goal when they had already been placed elsewhere in the map, they would be moved to the current location. This parameterization allows the adversary to place as many blocks as there are squares in the map. However, we found that adversaries trained with this parameterization drastically underperformed the alternative version used in the paper, scoring an average solved path length of $\approx 2$, as opposed to $\approx 15$. We hypothesize this is because when the adversary is randomly initialized, sampling from its random policy is more likely to produce impossible environments. This makes it impossible for the agents to learn, and cannot provide a regret signal to the adversary to allow it to improve. We suggest that when designing an adversary parameterization, it may be important to ensure that sampling from a random adversary policy can produce feasible environments.

The environment-constructing adversary's observations consist of a $15 \times 15 \times 3$ image of the state of the environment, an integer $t$ representing the current timestep, and a random vector $z \sim \mathcal{N}(0, I), z \in \mathbb{R}^{50}$ to allow it to generate random mazes. Because the sequencing of actions is important, we experiment with using an RNN to parameterize the adversary as well, although we find it is not always necessary. The adversary architecture is similar to that of the agents; it consists of a single convolutional layer which connects to an LSTM, and then to two fully connected layers which connect to the policy outputs. Additional inputs such as $t$ and $z$ are connected directly to the LSTM layer. We use a second, identical network to estimate the value function. To find the best hyperparameters for PAIRED and the baselines, we perform a grid search over the number of convolution filters used by the adversary, the degree of entropy regularization, which of the two parameterizations to use, and whether or not to use an RNN, and finally, the number of steps the protagonist is given to find the goal (we reasoned that lowering the protagonist episode length relative to the antagonist length would make the regret a less noisy signal). These parameters are shared between PAIRED and the baseline Minimax adversary, and we sweep all of these parameters equally for both. For PAIRED, we also experiment with whether the protagonist and antagonist optimize regret, and with using a non-negative regret signal, *i.e.* $\text{REGRET} = \max(0, \text{REGRET})$, reasoning this could also lead to a less noisy reward signal.

For the complexity experiments, we chose the parameters that resulted in the highest solved path length in the last 20% of training steps. The best parameters for PAIRED were an adversary with 128 convolution filters, entropy regularization coefficient of 0.0, protagonist episode length of 250, non-negative regret, and the agents did not optimize regret. The best parameters for the minimax adversary were 256 convolution filters, entropy regularization of 0.0, and episode length of 250. For the Population-based-training (PBT) Minimax experiment, the best parameters were an adversary population of size 3, and an agent population of size 3. All adversaries used a convolution kernel size

of 3, an LSTM size of 256, two fully connected layers of size 32 each, and a fully connected layer of size 10 that inputs the timestep and connects to the LSTM.

For the transfer experiments, we first limited the hyperparameters we searched based on those which produced the highest adversary reward, reasoning that these were experiments in which the optimization objective was achieved most effectively. We tested a limited number of hyperparameter settings on a set of validation environments, consisting of a different maze and labyrinth, mazes with 5 blocks and 40 blocks, and a nine rooms environment. We then tested these parameters on novel test environments to produce the scores in the paper. The best parameters for the PAIRED adversary were found to be 64 convolution filters, entropy regularization of 0.1, no RNN, non-negative regret, and having the agents themselves optimize regret. The best parameters for the minimax adversary in this experiment were found to be the same: 64 convolution filters, entropy regularization of 0.1, and no RNN.

## F.2 Hopper Experiments

The hopper experiments in this paper are in the standard MuJoCo [42] simulator. The adversary is allowed to apply additional torques to the joints of the agent at a some proportion of the original agent's strength $\alpha$. The torques that are applied at each time step are chosen by the adversary independent of the state of the environment, and in the PAIRED experiments, both the protagonist and the antagonist are given the same torques.

The adversaries observation consists of only of the the time step. Each policy is a DNN with two hidden layers each with width 32, $tanh$ activation internally and a linear activation on the output layer. They were trained simultaneously using PPO [35] and an schedule in the adversary strength parameter $\alpha$ is scaled from 0.1 to 1.0 over the course of 300 iterations, after which training continues at full strength. We pre-train agents without any adversary for 100 iterations.

Hyperparameter tuning was conducted over the learning rate values [5e-3, 5e-4, 5e-5] and the GAE lambda values $[0.5, 0.9]$. The minimax adversary worked best with a leaning rate of 5e-4 and a lambda value of $0.5$; PAIRED worked best with a leaning rate of 5e-3 and a lambda value of $0.09$. Otherwise we use the standard hyperparameters in Ray 0.8.