[Reviews · NeurIPS 2020]

Review 1

Summary and Contributions: This work introduces Unsupervised Environment Design as a paradigm for automatically learning a curriculum over environments for learning behaviorally complex policies that are robust and transfer well. UED implements an approximation of minimax regret, which aims at learning a policy with good worst-case *regret*. This is cast as an adversarial game. The paper presents a theoretical context for motivating this approach, an algorithmic approximation, and empirical demonstrations of its properties.

Strengths: The motivation for the proposed approach is well founded. In particular, minimax regret is compared with domain randomization and a minimax-style algorithm conceptually and experimentally. The work includes empirical evidence that their method enables zero-shot transfer to out-of-distribution and complex environments in their partially observable maze testbed environment. The work also includes some empirical evaluation of the claims that their algorithm creates a curriculum of complexity. A continuous control task is included to show more general applicability of the idea. The topics and techniques explored in this work represent active areas of research in the field and should be of interest to members of the NeurIPS community.

Weaknesses: The value added by the presented theoretical characterization is unclear. This is due, in part, to the way in which it is presented -- it is split between the main text and the appendix with proofs that are difficult to follow. The proposed method relies on using a separate policy (the antagonist) to estimate the regret of the primary policy (the protagonist). This should clearly produce systematically incorrect estimates of the regret, since there is little reason to expect the antagonist policy to be more optimal than the protagonist policy. This is regarded as a purposeful choice to induce a curriculum of learning. Still, any characterization of the learning dynamics between the two policies is entirely absent. This seems like an important omission since it is likely quite important to understand the tension between estimating regret and maintaining a good curriculum. There are potentially many choices for how to differentially train the protagonist/antagonist policies and the discussion of those choices is lacking.

Correctness: The claims and methodology do not raise any major concerns.

Clarity: For the most part. However, the material in the appendix is very difficult to follow. Considering that several portions of the main text are not self-contained (requiring the appendix to appreciate their significance), this is an issue. The appendix should be rewritten for clarity and the authors should consider reorganizing the theoretical content throughout.

Relation to Prior Work: Yes, it would seem so, though it is possible there are relevant previous works that I am not familiar with.

Reproducibility: Yes

Additional Feedback: I would be willing to raise my score if the authors can address my concerns about the tension between estimating regret and maintaining a suitable curriculum. Extra experiments, some empirical analysis, or a discussion of the practical considerations would go a long way. Minor issues: - The figures are very difficult to read and at times are missing important labels (Figure 4). Increase font size. - Typo in line 207. - Figure 3 is mislabeled (Labyrinth vs. Lava). - Typo in the legend for Table 2. - Also, tables and figures should be self-contained, but Table 2 requires the appendix to understand. - Same comment for theorems/proofs. Splitting them between the main text and appendix is not helpful. POST-REBUTTAL COMMENTS: The authors are running experiments to address my main concerns, so I am raising my score accordingly. Based on the other reviewers' scores, this seems likely to be accepted. So, I encourage the authors to provide as much meaningful analysis as possible regarding the practical choices underlying PAIRED -- in particular, choices of how to train the protagonist/antagonist. I expect these choices will matter, potentially in subtle ways, and it would be very helpful to offer readers guidance concerning these choices. Hopefully, the authors can provide a thorough discussion around this topic in the appendix.


Review 2

Summary and Contributions: This paper introduces a new paradigm for automatic curriculum learning on increasingly complex environments. The most significant contribution of this work is the PAIRED algorithm, which is a minimax framework that maximises the environment complexity (the algorithm generates the training environment) for the main learning agent while constraining to achievable scenarios.

Strengths: The good empirical results of this work suggests that this is a good line of work for automatic curriculum learning. It is also remarkable the novelty of the "alliance" of the "adversary" and the "antagonist" agents against the main learner (the protagonist). The goal of this alliance is to generate environments that are challenging to the protagonist while still feasible for the antagonist.

Weaknesses: The proofs of the theoretical claims need further work and clarification. The algorithm seems to work only when the adversary and antagonist policies successfully coordinate.

Correctness: Generally, the empirical work seems to be correct, however I can't say the same about the theoretical claims since their proofs are not clear enough in some points. The proof of the second theorem says that it hold while the adversary and antagonist policy successfully coordinate, how often does this happen? what training conditions are needed for this to happen? Tables in lines 498 and 503 have no caption, additionally, what is exactly Success and Failure in those tables?

Clarity: The paper is generally clear. However, the proofs of the theoretical claims of this paper need further revision. Note that, while these proofs appear in the appendix only, they are the ground of the main theoretical claims of this work. Some examples: *Proof of Theorem 1 is referenced before REGRET function is defined, and it is not defined in the proof either, what is the i in \pi_i referring to? why the superscript SAFE? *Theorem 1, could you further explain what do you mean when you say "that the range of possible rewards in either class is smaller than the difference between the classes"? *Proof of Theorem 2, what is the N superscript of the trajectory? Additionally, I think clarity for UED could be improved, specifically in line 143, I missed a clearer description saying we have X and want to achieve Y. As it is described, it's not totally clear. For instance, " to generate a distribution of environments in which to continue training a policy" then for this problem we already have a policy? when do we know when to stop?. Also "We would like to curate this distribution of environments to best support the continued learning of a particular policy" to support the continued learning of a policy with which goal? As a suggestion, I would give an specific example there of input goal and how the proposed approach using continued learning achieves it.

Relation to Prior Work: Yes, authors seem to have included a complete explanation on how their work differs from previous solutions

Reproducibility: Yes

Additional Feedback: Could you add a specific example/ problem that would be easily solved by defining it as a UED? I think it would help the paper in general. The agent for the Lava environment, that replaces walls with dangerous lava, is trained from generated maps with lava instead of walls? Additionally, the paper needs further proof reading, some minor mistakes I found: *Line 511: I wouldn't start a proof section saying "it would be nice to know that..." that is too informal *Line 512: "their" should be "its" *Line 40: Section ?, i.e., referenced section is missing the number *Line 138: the function T^M shouldn't be defined on S^M? *Line 171: This sentence needs further explanation *Line 207: based on twice *Line 209: Figure ? The Broader Impact section, specially the first paragraph is too speculative, automating jobs or automated weapons are general problems of the AI field, it should focus more on the impact of this specific work. For instance can we claim that UED achieves a robust agent? can we find ways for UED to cover corner cases that may happen in the real world? I think the authors have a good proposal and findings, but some some further work is still required so that this becomes a sound and exciting work to be accepted. ----- POST-REBUTTAL COMMENTS: I think authors have correctly addressed all my concerns with the theoretical proofs. I think the additional explanations and examples will greatly benefit the paper to have a higher impact. As I expressed in my first review I believe this is an interesting and exciting contribution. Therefore, I have raised my score to suggest the acceptance of this work.


Review 3

Summary and Contributions: This paper proposes a novel method to automatically generate distributions over valid, solvable environments. The authors formulate the task as a three-player minimax adversarial training game, where an protagonist agent is engaged to maximize the regret of generation policy, whereas an adversary agent and an antagonist agent are trained to maximize the regret. The authors provide theoretical analysis over the minimax regret formulation in appendix. Empirical evaluation demonstrates that the proposed method leads to promising zero-shot generalization performance.

Strengths: + Clearly written and easy to follow. + The idea of formulating the task as a three-player minimax adversarial training game is novel compared to the previous works. + Theoretical support of the minimax regret formulation + Throughout evaluation on the method on both grid-based navigation domain and a continuous domain from Mujoco. + Significant performance improvement over all the baselines on the two challenging zero-shot transfer tasks (labyrinth and maze)

Weaknesses: - Lack of discussion on scenarios where the method is not applicable - Lack of detailed analysis on the evaluation results

Correctness: The formulations are correct.

Clarity: The paper is well structured and easy to follow.

Relation to Prior Work: The discussion on related work is quite extensive.

Reproducibility: Yes

Additional Feedback:


Review 4

Summary and Contributions: The authors propose a framework for the unsupervised generation of environments and present the PAIRED algorithm which provides a novel approach to curriculum development and training agents capable of transfer and zero-shot learning. They relate environment design to decision theory and highlight interesting parallels between the two. They compare performance of agents trained under PAIRED, Domain randomization (DR), minimax training (MMT), and Population-Based Training minimax (PBT). In a small grid world PAIRED agents outperform DR and MMT on zero-shot transfer to established complex forms (Labyrinth and a length Maze design). The environments generated by PAIRED demonstrate a generally more stable curriculum of increasing complexity compared to DR, MMT, and PBT. In a continuous control task (a modification of MuJoCo hopper) they show similar results.

Strengths: The PAIRED algorithm provides an elegant solution to the problem of generating complex and challenging-yet-solvable environments. The empirical results demonstrate the interesting potential of this approach for creating robust agents that learn transferrable policies. The insight that pairing the adversarial environment design with an "antagonist" agent constrains the environment generated fully in difficult-but-achievable configurations is profound and could have significant impact on general RL research. Relating the issues of environment design in RL to established decision theory paradigms is also valuable and worthy of further study. The impact statement was thoughtful and considered.

Weaknesses: I was initially sceptical about the need to formalize unsupervised environment design (presented as UED) and still feel the authors could do more to situate UED with respect to RL literature. However the relation to decision theory is both clear and valuable. I am intrigued by the sequence of environments generated by the PAIRED algorithm and would have liked to see further exploration of the implications of that curriculum, uncoupled from the performance of the "protagonist" agent. Could this be used as a sequence of training tasks for a broad range of agents that are not specifically minimizing the regret w.r.t. the antagonist's performance? Is the decoupling possible? It would be good to provide a sketch of the proof of Theorem 2 in the body of the paper itself rather than relying entirely on the appendix.

Correctness: Theoretical results are supported mainly in the appendix but the tie to decision theory is sound. The empirical methodology is reasonable (of course more comparisons can always be demanded but I felt the results shown were sufficient to support the claims).

Clarity: The paper is clear and easy to follow.

Relation to Prior Work: The authors do an excellent job of relating the problem of environment design to established problems in decision theory, i.e. decisions under ignorance, the principle of insufficient reason, and minimax reason/regret. They also situated PAIRED in the RL research by contrasting their approach

Reproducibility: Yes

Additional Feedback:

[Author Response · NeurIPS 2020]

We thank all of the reviewers for their valuable feedback. We appreciate that **Reviewer 4** finds the theoretical connection to decision theory both "sound" and "clear and valuable" and that **Reviewer 3** states that "the formulations are correct." However, we acknowledge that the clarity and organization of the theory could be improved. To address this, we will explain the major theoretical results more clearly in the main body of the text, and include a sketch of the proof of Theorem 2 as suggested by **Reviewer 4**. We will incorporating **Reviewer 2**'s suggestions for improving the clarity of the goals and assumptions of the theoretical results. We will also provide additional experiments in the appendix, which test new methods for computing regret (for **Reviewer 1)** and assess whether a curriculum can be generated without training the protagonist using regret (for **Reviewer 4)**.

**Reviewer 2** points out that Therorem 2 shows that PAIRED recovers the minimax regret policy when the adversary and the antagonist successfully coordinate, but it is unclear this condition holds. However, coordination is not required for PAIRED to work. If the policies reach a Nash Equilibrium, the protagonist performs at least as well as the antagonist in every parameterization, since the protagonist could guarantee 0 payoff by using the antagonist's strategy and otherwise the adversary would choose a distribution on which the protagonist performs relatively worse. With a capable antagonist the protagonist would learn the minimax regret policy, even without the adversary and the antagonist coordinating. We will add this explanation to the paper.

**Reviewer 2** asks for clarification of the conditions of Theorem 1: there are two outcomes, Success and Failure, and the range of rewards given for Success does not overlap with the range of rewards given for Failure. Specifically, Successful outcomes give rewards in some range $[\mathbf{S}_{min}, \mathbf{S}_{max}]$, and Failure outcomes give rewards some in range $[\mathbf{F}_{min}, \mathbf{F}_{max}]$ such that $\mathbf{F}_{min} \leq \mathbf{F}_{max} < \mathbf{S}_{min} \leq \mathbf{S}_{max}$, and $\mathbf{S}_{max} - \mathbf{S}_{min} < \mathbf{S}_{min} - \mathbf{F}_{max}$ and $\mathbf{F}_{max} - \mathbf{F}_{min} < \mathbf{S}_{min} - \mathbf{F}_{max}$. We thank **Reviewer 2** for bringing this ambiguity to our attention and will update the paper accordingly.

**Reviewer 2** also offered the valuable advice to improve clarity by offering a "specific example/ problem that would be easily solved by defining it as a UED". One example is training a robot in simulation to pick up objects from a bin in a real-world warehouse. There are many possible configurations of objects, including objects being stacked on top of each other. We may not know *a priori* the typical arrangement of the objects, but can naturally describe them as parameterizations of a simulated environment. We can provide a curriculum of training configurations with either domain randomization, minimax, or PAIRED. We will introduce this example in the introduction and reference it throughout the formal introduction of UED to improve the clarity of the formalism.

**Reviewer 2** points out that "clarity for UED could be improved" with "a clearer description saying we have X and want to achieve Y". In our setting we are given a class of training environments, and our goal is to construct a policy which performs well across a large set of these environments. We evaluate with a set of specific environments used as transfer tasks. To train such a policy, we start with an initially random policy, generate environments which are tuned to help it learn, train the policy on those environments, and repeat until convergence or until we have exceeded available computational resources. We call the problem of choosing how to generate these environments UED. The choice of how to solve the UED problem affects both the curriculum it generates and how the policy prioritizes performance in different environments at convergence. We will add phrasing to this effect in the introduction of the paper.

**Reviewer 1** points out that although using the 'relative regret' between the protagonist and an imperfect antagonist leads to an effective curriculum, it may be an inaccurate estimate of the worse-case regret during training. They add, "There are potentially many choices for how to differentially train the protagonist/antagonist policies". We are conducting additional experiments empirically investigating alternative methods for estimating regret: 1) we use a population of agents and compute regret as the difference between the highest performing agent and the population average, 2) we relabel the the current best-performing agent as the antagonist. We will include the results of these additional experiments in the appendix.

**Reviewer 4** asked whether it is possible to decouple generating a curriculum of environments using regret, and having the agent itself optimize regret. We have experimented with training the protagonist with environmental reward and not regret and find that it still performs well; we will provide these results in the appendix. In addition, the performance of PAIRED in the random 50 blocks environment indicates that it would perform well as a policy for the domain randomization setting, even though it was designed for a different objective.

**Reviewer 3** - we will provide further discussion on scenarios where the method is not applicable, including settings in which you already have an accurate model of the test environments you expect to encounter.

We will improve the Broader Impact statement according to the suggestions of **Reviewer 2**, making it more specific and focusing on PAIRED's ability to train more robust agents that "cover corner cases that may happen in the real world". We will correct all typos identified by the reviewers, including the caption of Figure 3 which mistakenly refers to a Lava environment. We appreciate your detailed feedback.

[Meta-Review · NeurIPS 2020]

This paper pursues a significant line of enquiry regarding an important topic: automatic, unsupervised environment design. The paper makes algorithmic, theoretical, and empirical contributions. While the reviewers had some concerns about the clarity of the theory and the adequacy of the empirical results, these have been well addressed in the rebuttal. The authors are strongly urged to incorporate all the reviewers' feedback in the final version.